# Oncogenic Role of Tumor Necrosis Factor α-Induced Protein 8 (TNFAIP8)

**DOI:** 10.3390/cells8010009

**Published:** 2018-12-24

**Authors:** Suryakant Niture, Xialan Dong, Elena Arthur, Uchechukwu Chimeh, Samiksha S. Niture, Weifan Zheng, Deepak Kumar

**Affiliations:** 1Julius L. Chambers Biomedical Biotechnology Research Institute (BBRI), North Carolina Central University, Durham, NC 27707, USA; sniture@nccu.edu (S.N.); earthur1@nccu.edu (E.A.); uchimeh@nccu.edu (U.C.); 2Bio-manufacturing Research Institute and Technology Enterprise (BRITE), North Carolina Central University, Durham, NC 27707, USA; xdong@nccu.edu (X.D.); wzheng@nccu.edu (W.Z.); 3Catonsville High School, Catonsville, MD 21228, USA; samniture@gmail.com; 4Department of Pharmaceutical Sciences, North Carolina Central University, Durham, NC 27707, USA

**Keywords:** tumor necrosis factor α (TNFα), tumor necrosis factor α-inducing protein 8 (TNFAIP8), oncogene, autophagy, cell survival

## Abstract

Tumor necrosis factor (TNF)-α-induced protein 8 (TNFAIP8) is a founding member of the TIPE family, which also includes TNFAIP8-like 1 (TIPE1), TNFAIP8-like 2 (TIPE2), and TNFAIP8-like 3 (TIPE3) proteins. Expression of TNFAIP8 is strongly associated with the development of various cancers including cancer of the prostate, liver, lung, breast, colon, esophagus, ovary, cervix, pancreas, and others. In human cancers, TNFAIP8 promotes cell proliferation, invasion, metastasis, drug resistance, autophagy, and tumorigenesis by inhibition of cell apoptosis. In order to better understand the molecular aspects, biological functions, and potential roles of TNFAIP8 in carcinogenesis, in this review, we focused on the expression, regulation, structural aspects, modifications/interactions, and oncogenic role of TNFAIP8 proteins in human cancers.

## 1. Introduction

Chronic inflammation is one of the causative factors in cancer development [1]. Cellular inflammation can occur by various factors including pathogenic microbial infections, consumption of drugs/alcohol, or exposure to various environmental toxins. Cellular mechanisms reveal that chronic inflammation activates pro-inflammatory pathways, cytokines, and chemokines which are involved in the development of cancer risk [2,3]. Dysregulation of the production of cytokine tumor necrosis factor α (TNFα) in the cell is known to be associated with the development of various human diseases including Alzheimer’s disease [4], major depression [5], inflammatory bowel disease [6], disorders of glucose and lipid metabolism [7], and cancer [8,9]. Protein factors/bio-molecules which are associated with the neutralization of TNFα-mediated inflammation and apoptosis play an important role in the treatment of cancer because of their association with drug resistance [10]. Importantly, TNFα regulates and controls the expression of the tumor necrosis factor-α-induced protein 8-like (TNFAIP8/TIPE) family of proteins, including tumor necrosis factor (TNF)-α- induced protein 8 (TNFAIP8) and TNFAIP8-like 2 (TIPE2). Several reports suggest that TNFAIP8 and TNFAIP8-like 3 (TIPE3) proteins promote cell survival and drug resistance [11,12], whereas TNFAIP8-like 1 (TIPE1) and TIPE2 have been implicated in cellular apoptosis [13,14]. TIPE family proteins are structurally similar and show ~54% of amino acid sequence identity and ~75% of amino acid sequence similarity to each other (Figure 1A), however, all members play highly diverse and distinct roles in various human cancers.

TNFAIP8, also called SCCS2, GG2-1, MDC-3.13, NDED, and SSC-S2, is the first TIPE family protein [11,15,16]. Multiple transcripts of TNFAIP8 and five protein isoforms of TNFAIP8 have been reported (https://useast.ensembl.org/index.html, accessed on: 2 October 2018). The molecular weights of these isoforms range between 18 kDa and 22 kDa with highly conserved C-terminal amino acid sequence homology and a slight variation at the N-terminal. Although TNFAIP8 isoform 2 is predominantly expressed in several cancers [17,18], the roles of individual TNFAIP8 isoforms are not clear so far and therefore, in the current review, we focused on the regulation, structure, interactions, and biological functions of TNFAIP8 proteins in human cancer.

## 2. TIPE Family Proteins: TIPE1, TIPE2, and TIPE3 and Their Brief Roles in Human Disease

TNFα is a cell signaling cytokine, which regulates cellular inflammation and modulates inflammatory diseases [19]. TNFα binds with TNFR1 (TNF receptor type 1) and TNFR2 (TNF receptor type 2) [20], activates the nuclear factor κB (NF-κB) pathway, and induces expression of TIPE family proteins including TNFAIP8 and TIPE2 [17,21]. All TIPE family members are mostly localized in the cytosol, and nuclear localization of TNFAIP8 is also reported [22]. The TIPE family of proteins consists of four members: TNFAIP8, TIPE1, TIPE2, and TIPE3. TIPE1 shares 55.9%, TIPE2 shares 52.7%, and TIPE3 isoform 2 shares 56.4% amino acid identity with TNFAIP8 isoform 2. In addition, TIPE1 shares 78.7%, TIPE2 shares 75.5%, and TIPE3 isoform 2 shares 80.9% amino acid sequence similarity with TNFAIP8 isoform 2 (Figure 1A, http://www.ebi.ac.uk/Tools/psa/emboss_needle/, accessed on: 10 October 2018) [23]. Interestingly, mouse TNFAIP8 shares 89.4% amino acid sequence identity and 94.4% amino acid sequence similarity with human TNFAIP8 (Figure 1A). Amino acid sequence alignment of TIPE family members also showed highly conserved residues at C-terminus and significant variation in the N-terminus (Figure 1B) (https://www.ebi.ac.uk/Tools/msa/clustalo/, accessed on: 5 October 2018). The biological roles of TIPE1, TIPE2, and TIPE3 in human diseases are discussed below.

**TNFAIP8-like 1 (TIPE1)**: TIPE1 (Oxi-β) is expressed in neurons, hepatocytes, muscle tissues, and germ cells, but not in mature B and T cells [24]. High expression of *TIPE1* mRNA is also present in many cancer cell lines including myeloid HMC-1, sarcoma U-2OS, skin cancer A-431, and liver cancer HepG2 [25] (https://www.proteinatlas.org/ENSG00000185361-TNFAIP8L1/cell#rna, accessed on: 9 September 2018). In hepatocellular carcinoma, TIPE1 is downregulated and associated with Tumor Node Metastasis (TNM) staging and patient death [13]. TIPE1 interacts with Rac1, inhibits p65 and c-Jun N-terminal kinase, and induces caspase-mediated cellular apoptosis in liver cancer cells [13]. Oxidative stress is known to induce TIPE1 expression, leading to mammalian target of rapamycin (mTOR) inhibition, which results in cellular autophagy and cell death in neuronal cell lines [26]. Overexpression of TIPE1 stabilizes tuberous sclerosis complex 2 (TSC2) by competing for the binding of TSC2 with F-box/WD repeat-containing protein 5 (FBXW5). TIPE1 stabilizes tuberous sclerosis complex 2 (TSC2), a negative regulator of mTOR signaling, and thus, TIPE1 facilitates cell death [26]. Ectopic overexpression of TIPE1 in lung cancer cells reduces cell colony formation and proliferation and induces apoptosis by regulation of cyclin D1, cyclin B1, caspase 8, caspase3, matrix metallopeptidase 2 (MMP2), and matrix metallopeptidase 9 (MMP9) expression. Similarly, in a homograft tumor model in Balb/c mice, TIPE1 expression inhibited the tumor growth and reduced the tumor weight of murine lung cancer homografts, suggesting that TIPE1 acts as an anti-tumor molecule in lung cancer [27]. Indeed, TIPE1 induces cell apoptosis and inhibits tumorigenesis.

**TNFAIP8-like 2 (TIPE2)**: TIPE2 is another member of TIPE family which has been highly studied so far, and its role is mostly associated with dysregulation of immunity and inflammation. Studies demonstrate that TIPE2 is mostly expressed in bone marrow and in the immune system, and overall is a negative regulator of immunity and inflammation. *TIPE2* knockdown in mice induces multi-organ inflammation and premature death [21]. *TIPE2*-knockdown mice become hypersensitive to toll-like receptor (TLR) stimulation [21]. The expression of *TIPE2* mRNA/protein and its co-relation to different human diseases is well documented. Reduction of *TIPE2* mRNA levels has been observed in peripheral blood monocytes (PMBCs) in human patients with systemic lupus erythematosus [28], childhood asthma, and myasthenia gravis [29]. Downregulation of TIPE2 increases the levels of interleukin-6 (IL-6), interleukin-17 (IL-17), and interleukin-21 (IL-21) in these human diseases [29]. TIPE2 expression also modulates chronic hepatitis B virus infection. Down-regulation of TIPE2 is negatively associated with viral load and serum markers of liver inflammation [30]. A decreased expression of TIPE2 has been observed in the PBMCs of patients with chronic hepatitis B infection [30] and patients with primary biliary sclerosis [31]. In myocardial ischemia/reperfusion injury, TIPE2 inhibits nucleotide-binding oligomerization domain-containing protein 2 (NOD2), activates mitogen-activated protein kinases MAPK and NF-κB signaling, and negatively regulates NOD2-mediated inflammatory signaling [32].

In human cancer, decreased expression of TIPE2 is observed in hepatic cancer [33], gastric cancer tissues [34] and small cell lung cancer [35]. On the other hand, a positive co-relation of TNM staging with increased expression of TIPE2 is observed in renal cell carcinoma [36]. Similarly, in colon cancer tissues, TIPE2 expression is positively associated with lymph node metastases and the Duke stage of cancer [37]. *TIPE2* knockdown activates Ral and AKT (protein kinase B), increases resistance to cell death, increases migration, and dysregulates exocyst complex formation. On the other hand, overexpression of TIPE2 induces cell death and inhibits Ras-induced tumorigenesis in mice. [33]. Increasing TIPE2 expression decreases cell proliferation by upregulating N-ras and p27 expression in gastric cell lines [34]. TIPE2 also regulates AKT and extracellular signal-regulated kinase 1/2 (ERK1/2) signaling. Adenovirus-directed expression of TIPE2 suppresses gastric cancer growth by induction of apoptosis and inhibition of AKT and ERK1/2 signaling [38], suggesting that, similar to TIPE1, TIPE2 mostly inhibits various cancer cell growths by the induction of apoptosis.

**TNFAIP8-like 3 (TIPE3)**: The biological role of TIPE3 is still unknown; only a few studies have shown that TIPE3 is an oncogenic molecule and that increased levels of TIPE3 are present in cervical, colon, lung and esophageal cancers [39,40]. TIPE3 regulates PI3K/AKT signaling, and knocking down *TIPE3* reduces tumor development in animals [39]. TIPE3 protein promotes breast cancer metastasis by activating AKT and NF-κB signaling pathways [41], suggesting that TIPE3 may be involved in cancer cell survival.

By introducing and providing an overview of the functional roles of TIPE1, TIPE2, and TIPE3 proteins, we have laid the foundation to discuss the focal point of this review: the molecular, structural, and functional roles of the founder member of TIPE family protein—tumor necrosis Factor α-inducing protein 8 (TNFAIP8).

## 3. Tumor Necrosis Factor α-Inducing Protein 8 (TNFAIP8)

### 3.1. TNFAIP8 Expression and Regulation 

TNFAIP8 proteins were first identified by comparing two primary and matched metastatic head and neck squamous cell carcinoma cell lines [15], as well as a TNFα-inducible gene in endothelial cells [16]. Expression of TNFAIP8 is reported in most human tissues; however, the relative mRNA expression is not consistent with protein expression in many human organs. Recent human Protein Atlas data clearly suggest that *TNFAIP8* mRNA and protein expression are mostly found in the bone marrow and immune system, gastrointestinal tract, lung, and adipose tissue [25] (http://www.proteinatlas.org/ENSG00000145779-TNFAIP8/tissue, accessed on: 25 September 2018). Moderate levels of mRNA expression are found in male epididymis, seminal vesicle, testis and prostate tissues. In the female, TNFAIP8 mRNA and protein expression are found in the fallopian tube, cervix, and uterine endometrium. Lower levels of mRNA and undetectable levels of protein expression are found in many human tissues such as the pancreas, salivary and thyroid glands, kidney, liver, and ovary.

The human *TNFAIP8* gene is localized at chromosome 5 in the forward strand q23 region, and so far, eight transcripts (splice variants) are reported (https://www.ncbi.nlm.nih.gov/nuccore/?term=human+tnfaip8 accessed on: 3 October 2018). TNFAIP8 expression in cells is regulated by several factors including transcription factors, NF-κB, Hif, and chicken ovalbumin upstream promoter transcription factor I (COUP-TFI) [12,22,42,43]. NF-κB induces the expression of TNFAIP8, which leads to increased cell survival [42]. Expression of TNFAIP8 is also controlled by TNFα [17,22]. TNFα can bind to tumor necrosis factor receptor 1/2 (TNFR1/TNFR2) and induce cellular inflammation leads to the dissociation of nuclear factor κB (NF-κB) from its inhibitor IκBα and subsequently, activation and nuclear localization of NF-κB. NF-κB may bind with the *TNFAIP8* promoter and induce the expression of TNFAIP8 [17,42]. In response to androgen, an induction of TNFAIP8 protein is reported in prostate cancer cells [17,22]. Moreover, genome-wide and gene ontology analyses indicate an altered expression of *TNFAIP8* in long-term androgen-deprived LNCaP AI cell line, suggesting that androgen receptor (AR) may modulate expression of TNFAIP8 [44]. On the other hand, expression of the TNFAIP8 protein is suppressed in Hela cells by chicken ovalbumin upstream promoter transcription factor I (COUP-TFI) and DBC1 (deleted in bladder cancer protein 1), a pro-apoptotic protein complex [12,43]. TNFAIP8 protein expression is also controlled by promoter methylation in prostate epithelial cancer cells [45]. Hypomethylation of the *TNFAIP8* gene increases the expression of the TNFAIP8 protein in the placenta and peripheral blood cells from early-onset pre-eclamptic patients. The study suggests that *TNFAIP8* gene methylation may be associated with the pathogenesis of pre-eclampsia [46]. In addition, variations in DNA methylation of the *TNFAIP8* gene are also associated with the pattern of birth rate in the black population [47], suggesting that TNFAIP8 expression is controlled not only by transcription factors but also by promoter methylation of the *TNFAIP8* gene.

### 3.2. TNFAIP8 Structure

Although the *hTNFAIP8* gene encodes eight transcripts (variants), only five protein variants/isoforms have been reported so far (https://www.ncbi.nlm.nih.gov/nuccore/?term=human+tnfaip8, accessed on: 5 October 2018). *TNFAIP8* transcript 1 codes for 198 amino acids, transcript 2 codes for 188 amino acids, transcript 4 codes for 210 amino acids, transcript 5 codes for 166 amino acids, and transcript 7 codes for 201 amino acids (Figure 2A). *TNFAIP8* transcripts 1 and 3 code for the same amino acid sequences (198 amino acids), and similarly, transcripts 5, 6, and 8 also code for the same amino acid sequences (166 amino acids), but their 5′ noncoding transcript regions are highly variable. Amino acid alignments of these isoforms clearly show highly conserved sequences at their C-terminal region in all isoforms and variable amino acid sequences at the N-terminal regions between isoforms 1, 2, 4 and 7 (Figure 2B) (https://www.ebi.ac.uk/Tools/msa/clustalo/ accessed on: 5 October 2018). Domain mapping reveals that TNFAIP8 proteins contain a highly conserved coiled-coil structural motif, which is present in all TNFAIP8 isoforms (Figure 2B,C). The coiled-coil structural motif sequence is highly conserved in hTNFAIP8 and mTNFAIP8 proteins but shows significant variations in TIPE1, TIPE2, and TIPE3. Generally, the coiled-coil motif in the proteins consists of at least 2–7 α-helices, which are coiled together like the strands of a rope in dimer and trimer shapes [48,49]. Oncogenes c-Fos and c-Jun contain such coiled-coil structural motifs, and these motifs are involved in numerous biological functions [50]. In addition to the coiled-coil domain, TNFAIP8 proteins contain a destruction box (D-Box) consensus (RNVLSRLLN) between amino acids 115 and 123 in human TNFAIP8 (hTNFAIP8) isoform 2 (Figure 2C). The same D-Box consensus is also found in mTNFAIP8. Interestingly, 80% of the TIPE3 D-Box consensus sequence is similar to the hTNFAIP8 and mTNFAIP8, and a significant variation in TIPE1 and TIPE2 D-Box consensuses with hTNFAIP8 was observed (Figure 1B). The D-Box has been reported in cell cycle-related proteins like p21 [51], cyclin B1 [52], cyclin A2 [53], Nek2A [54] and STK31 [55] (Figure 2C). The D-Box consensus facilitates the degradation of cell cycle-related proteins by anaphase-promoting complex or cyclosome (APC/C) during cell cycle progression [56]. The biological significance of the D-Box consensus present in TNFAIP8 proteins is still unknown, but a recent study demonstrates that TNFAIP8 modulates cell cycle (S–phase) in liver cancer cells [57].

In an earlier study, we demonstrated that the amino acid sequence of TNFAIP8 (SCC-S2) also contained a putative death-effector domain (DED), which showed significant homology or similarities (more than 25%) with DED II domain of cFLIP (cell death regulatory proteins) and CASH (caspase 8/10 homologs) proteins [11]. The DED domain-containing proteins interact with death receptors such as tumor necrosis factor receptor 1 (TNFR1), leading to the recruitment of adaptor proteins such as TNFR-associated death domain (TRADD), TNF receptor-associated factor (TRAF), Fas-associated death domain (FADD), and caspase-8/FADD-like IL-1β converting enzyme (FLICE) [11]. Since TNFAIP8 possesses a DED domain at the N-terminus and lacks a caspase catalytic domain at the C-terminus, TNFAIP8 may act as an inhibitor for DED domain-associated protein-like FLICE [11]. Interestingly, TIPE2 crystal structure analysis revealed that the putative death-effector domain (DED) is not actually a DED domain, but a mirror image of DED in topological structure [58], suggesting that TIPE2 does not contain a DED domain but consists of a large hydrophobic deep central cavity.

Recently, Kim et al. [59] analyzed the crystal structure of mouse TNFAIP8 complex with phosphatidylethanolamine (TNFAIP8-PE). The overall structure shows a cylindrical domain with a large central cavity almost the same as that of human TIPE2 [58] and human TIPE3 [39], indicating that the TIPE family shares a common structural motif. The volume of the cavity of mTNFAIP8 is 837 Å, which is closer in size to hTIPE2 and hTIPE3. Interestingly, the central cavity is lined with highly conserved hydrophobic residues with a few differing residues such as Leu109 of mTNFAIP8, which is substituted with Gly97 and Thr203 in hTIPE2 and hTIPE3 respectively, and Leu51 of mTNFAIP8, which is replaced by Phe145 in hTIPE3 [59]. Since the human TNFAIP8 crystal structure is not solved yet, using a molecular modeling approach, we performed comparative structural analysis of hTNFAIP8 with mTNFAIP8. We used the hTNFAIP8 isoform 2 amino acid sequence (188 amino acids) to generate the predicted structure of hTNFAIP8 (Figure 3A,B). The hTNFAIP8 structure shows similar characteristics to mTNFAIP8 and comprises seven cylindrical helices (Figure 3A,B). The coiled-coil structural motif is localized at α helices 2 and 3 (Figure 3B left panel—labeled in green color) and the D-Box motif is present in α helix 5 (Figure 3B left panel—labeled in red color). The predicted structure of hTNFAIP8 shows a large cylindrical central hydrophobic cavity surrounded by seven cylindrical helices (Figure 3B—middle panel). The coiled-coil motif, which is present in α helices 2 and 3, is directly involved in the formation of a central cavity, whereas, the D-box consensus residues do not show any interaction with central cavity residues. The predicted structure of hTNFAIP8 shows a large central cylindrical cavity with a deep hydrophobic pocket. The mouth of the cavity possesses most of the hydrophilic residues (pink color), and highly hydrophobic residues are in the deep pockets (green color) (Figure 3B middle panel). The homology model of hTNFAIP8 with phosphatidylethanolamine (PE) clearly shows that PE binds with hTNFAIP8 similarly to mTNFAIP8 (Figure 3B, middle panel). Interestingly, when mTNFAIP8 was superimposed with hTNFAIP8, the structural data clearly suggested that in mTNFAIP8, His86-N formed a hydrogen bond with the phosphate group of PE, whereas in hTNFAIP8, Tyr76-O formed a hydrogen bond with the phosphate group of PE (Figure 3B right panel). The H-bond distance between the tyrosine-O in hTNFAIP8 and the phosphate-O of PE is 2.92 Å, and the distance between the His-N in mTNFAIP8 and phosphate-O of PE is 3.12 Å. The molecular modeling data clearly suggest that the hydrophilic mouth residues of the cavity may determine the ligand specificity/selectivity for binding with mTNFAIP8 or hTNFAIP8. Since this central cavity is conserved among TIPE family members, many similar hydrophobic cofactors or substrates are expected to bind inside the cavity and could be of therapeutic interest.

### 3.3. TNFAIP8 Interactions and Signaling

TNFAIP8 interacts with several proteins/factors and regulates cell signaling. In yeast, TNFAIP8 is found to interact with G alpha (i) coupled receptors and inhibits cell death in a caspase-independent manner in Balb-DS2S cells [60]. In earlier studies, it has been showed that TNFAIP8 may interact with GDNF (glial cell-derived neurotrophic factor) family receptor α1, Serine/arginine-rich splicing factor 2, PARP1, Importin α 1, DEAD box polypeptide 20, protein tyrosine phosphatase, and cyclin E1 [22]. Using co-immunoprecipitation assay followed by mass-spectrometric proteomic analysis, Chittaranjan et al. [60] demonstrated that *Drosophila* TNFAIP8 homolog protein CG4091/sigmar interacts with cytoskeletal proteins such as Pav, Map205, Act42, and α Tub84B, and modulate autophagy [61,62]. Previous work using high-throughput analysis of changes in the interactome suggests that TNFAIP8 interacts with autophagy-related protein 3 (ATG3), tyrosine 3-monooxygenase/tryptophan 5-monooxygenase activation protein epsilon (YWHAE), argininosuccinate synthase 1 (AAS1), protein phosphatase 1 regulatory (inhibitor) subunit 2 (PPP1R2), NHL repeat containing 2 (NHLRC2), importin 5, (IPO5), eukaryotic translation initiation factor 6 (EIF6), dipeptidyl-peptidase 3 (DDP3), dihydrolipoamide dehydrogenase (DLD), and acyl-CoA thioesterase 7 (ACOT7) [63]. Human interactome network analysis by Rolland et al. [64] suggested that TNFAIP8 may also interact with pleckstrin homology domain-containing family F member 2 (PLEKHF2), mediator complex subunit 4 (MED4), INO80 complex subunit E (INO80), TRAF-interacting protein with forkhead-associated domain (TIFA), proline-rich 13 (PRR13), and syndecan binding protein (SDCBP). Proteomic analysis using mass spectrometry reveals that TNFAIP8 may interact with protein tyrosine phosphatase type IVA member 1 (PTP4A1) [65]. The detailed TNFAIP8-interacting partners are presented in BioGRID (https://thebiogrid.org/117344, accessed on: 12 September 2018), however, it is not well established (1) whether all interacting partners physically interact with TNFAIP8, and (2) the biological significance of each interaction. Recently, using immunoprecipitation assay, we demonstrated that TNFAIP8 interacts with ATG3 and facilitates autophagy in prostate cancer cells [17]. In hepatocellular carcinoma, TNFAIP8 interacts with large tumor suppressor kinase 1 (LATS1), a serine/threonine-protein kinase, and overexpression of TNFAIP8 inhibits Yes-associated protein (YAP) phosphorylation, resulting in nuclear localization and stabilization of YAP. This leads to the upregulation of cell cycle-related proteins and cell proliferation. The study reveals that TNFAIP8 promotes liver cancer growth through LATS1-YAP signaling [57].

### 3.4. TNFAIP8 Modification

Currently, no post-translation modification of TNFAIP8 proteins has been reported. However, CBS Server (http://www.cbs.dtu.dk/services/, accessed on: 5 June 2018) predicted few amino acid modifications sites of the TNFAIP8 protein. We used TNFAIP8 isoform 2 (amino acids 188) amino acid sequence for prediction of post-translational modifications. The N-terminal acetylation sites were predicted using NetAcet 1.0 Server, C-mannosylation sites were predicted using NetC-Glyc, and N-linked glycosylation sites were predicted using NetNGlyc server. N-terminal acetylation prediction suggests that Ala2 and The3 may be possible sites for acetylation, however, acetylation scores were below 0.5 (non-significant). C-mannosylation or N-linked glycosylation sites were not predicted in TNFAIP8 protein. Interestingly, NetOGlyc 4.0 Server predicted TNFAIP8-The48 as an O-GAlNAc glycosylation site with a score of 0.54 (significant). In addition, we also analyzed predictable phosphorylation sites of TNFAIP8 protein using NetPhos 3.1 server (http://www.cbs.dtu.dk/services/NetPhos/, accessed on: 15 July 2018) and Prosite search (https://prosite.expasy.org/, accessed: 10 July 2018) (Figure 4A). The Netphos analysis revealed that TNFAIP8 can be phosphorylated by protein kinase A (PKA), protein kinase C (PKC), casein kinase II (CK II) and DNA-dependent protein kinase (DNAPK) (Figure 4B). The predictions suggest that kinase PKA may phosphorylate TNFAIP8 at S25 and S27; kinase PKC at T31, T68, T102, T112, and T138; kinase CKII at T36, S37, S38, S153; and DNAPK at T52. All putative kinases show NetPhos scores more than 0.5 (significant). Moreover, we used a Prosite search engine to analyze the possible kinase and phosphorylation sites in TNFAIP8. Similar to NetPhos predictions, the Pro-site server suggests that CKII may be involved in the phosphorylation of T31, T36, and S153 of TNFAIP8. The predictions also suggest that PKC may phosphorylate the T138 site of TNFAIP8. Interestingly, in TNFAIP8, no tyrosine phosphorylation site is predicted by either search engine. These data suggest that TNFAIP8 may be a phospho-protein, which may modulate cancer cell signaling directly or indirectly. Since TNFAIP8 possesses a D-Box motif, which is known to be involved in degradation of cell cycle-related proteins, and cyclin E1 is an interacting partner of TNFAIP8, it would be interesting to investigate the post-translational modifications of TNFAIP8 and their roles in cell cycle modulation and cell signaling.

## 4. Oncogenic Roles of TNFAIP8

Thus far, the data reviewed in the literature suggest that TNFAIP8 and TIPE3 may be involved in cell survival [11,12], whereas, TIPE1 and TIPE2 are involved in cellular apoptosis [13,14] (Figure 5A). The biological significance of TNFAIP8 protein in the regulation of cancer biology is reported in the literature with limited studies [66]. Initially, we showed that overexpression of TNFAIP8 in HeLa cells reduces apoptotic cell numbers as compared with the vector-transfected cells, suggesting that TNFAIP8 acts as an anti-apoptotic molecule [11]. We also showed that TNFAIP8 modulates breast cancer cell progression [67]. Overexpression of TNFAIP8 in MDA-MB-435 cells increases cell growth and tumorigenicity and enhances breast cancer cell migration by up-regulation of collagen I [67]. Similarly, overexpression of TNFAIP8 significantly increases the expression of vascular endothelial growth factor 2 (VEGFR-2), matrix metallopeptidase 1 (MMP-1), and matrix metallopeptidase-9 (MMP-9), and promotes MDA-MB-435 cell metastasis [68]. TNFAIP8 expression also correlates with ductal breast cancer (IDC). Patients with higher levels of TNFAIP8 developed tumors with higher-grade malignancy and had shorter survival times than patients with low TNFAIP8 levels [69].

In prostate cancer, TNFAIP8 is a potential biomarker [70], and depletion of TNFAIP8 increases expression of genes associated with anti-proliferation and apoptosis, for example, *IL24*, *FAT3*, *LPHN2*, and *EPHA3*. *TNFAIP8* depletion also increases the expression of fatty-acid oxidation gene, ACDL, and decreases the expression of several oncogenes such as *NFAT5*, *MALAT1*, *MET*, *FOXA1*, *KRAS*, *S100P*, and *OSTF1* [12]. Interestingly, increased nuclear localization of TNFAIP8 and its interaction with karyopherin alpha 2 in the nucleus is associated with a higher risk of prostate cancer recurrence [22]. Systemically administered liposome-entrapped TNFAIP8 antisense oligonucleotide (LE-AS5) decreases TNFAIP8 expression in PC3 prostate tumor-bearing athymic mice, which leads to tumor growth when mice were exposed to radiation or anticancer drug, docetaxel, suggesting that, TNFAIP8 may modulate radiation or drug-mediated resistance in prostate tumors [22]. Recently, using Microarray analysis, we have shown that expression of TNFAIP8 in PC3 prostate cancer cells downregulates the mRNA expression of several cell cycle-related genes, including *CCNB2*, *CCNE2*, *CDK2*, *CHEK*, and *PCNA*, however, no major cell cycle changes were observed. [17]. Interestingly, we also demonstrated that TNFAIP8-induced autophagy in prostate cancer and breast cancer cells leads to increased drug resistance and cell survival in prostate cancer cells [17].

In lung cancer, *TNFAIP8* mRNA levels are higher in cancer tissues compared with healthy lung tissue donors [71]. On the other hand, the data also demonstrated that *TNFAIP8* mRNA and TNFAIP8 protein levels are lower in lung tumor-infiltrating CD4+ and CD8+ T cells, compared with peripheral CD4+ and CD8+ T cells. In addition, in patients with advanced stages of lung cancer, the expression of TNFAIP8 in tumor-infiltrating CD8+ T cells is lower compared to patients with primary stages of lung cancer, suggesting that TNFAIP8 may be involved in the progression of non-small cell lung cancer (NSCLC) [71]. Previously, we demonstrated that antisense oligonucleotide-mediated inhibition of endogenous *TNFAIP8* decreases expression of VEGF receptor-2 in tumor cells and in human normal lung microvascular endothelial cells (HMVEC-L), and inactivation of *TNFAIP8* decreases the expression of metastasis-related molecules MMP-1 and MMP-9, suggesting that TNFAIP8 plays an important role in lung cancer tumor progression [68]. The molecular mechanistic role of TNFAIP8 in the regulation of Hippo signaling by interaction with LATS1 is reported in lung cancer cells [72]. TNFAIP8 interacts with LATS1 and decreases LATS1 phosphorylation, which leads to increased nuclear localization of YAP protein, resulting in increased lung cancer cell proliferation and invasion [72]. In lung cancer A549 cells, the expression of TNFAIP8 variant 2 (TNFAIP8-v2) inhibits p53 expression and decreased p53 binding to its target gene promoters [18]. However, *TNFAIP8-v2* knockdown increased the expression and binding of p53 to its target genes such as *CDKN1A*, *GADD45A*, *RRM2B*, and others, leading to increased expression of p21, cell cycle arrest, and doxorubicin-mediated DNA damage, clearly suggesting that TNFAIP8 controls p53 function and regulates lung cancer cell progression [18]. In addition, expression of p53K120R mutant (mutation of K120R of the DNA-binding domain of p53) in lung cancer *p53*-null H1299 cells induces expression of TNFAIP8. Mutant p53K120R binds the *TNFAIP8* locus at a cryptic *p53* response element that is not occupied or bound by wild-type p53. The data further suggest that the induction of TNFAIP8 by p53-K120R mutation increases lung cancer cell survival [73]. These studies indicate that TNFAIP8 negatively regulates apoptosis and promotes lung cancer cell growth.

The role of TNFAIP8 in liver cancer remains elusive, and only a few reports suggest that TNFAIP8 is involved in liver carcinogenesis. Recently, Dong et al [57] documented that expression of TNFAIP8 in liver cancer cells induces cell proliferation, migration, invasion, and xenograft tumor growth of hepatocellular carcinoma (HCC). Similar to lung cancer cells TNFAIP8 modulates the Hippo pathway in liver cancer cells by inhibition of YAP phosphorylation and by interaction with LATS1. A TNFAIP8–LATS1 interaction increases YAP nuclear localization and stabilization, resulting in upregulation of cell proliferation. Knockdown of LATS1 or YAP by siRNA blocked the effects of TNFAIP8 on cell proliferation, suggesting that TNFAIP8 promotes hepatocellular carcinoma progression through LATS1-YAP signaling pathway [57]. Beyond cancer, the biological role of TNFAIP8 in liver infection has been investigated and the study demonstrated that TNFAIP8 regulates *Listeria monocytogenes* infection by inhibiting Ras-related C3 botulinum toxin substrate 1 (RAC1). TNFAIP8-knockout mice are resistant to lethal *L. monocytogenes* infection and have a decreased bacterial load in the liver and spleen [74].

In gastric cancer, the microRNA-9 expression is lower compared to adjacent non-cancerous tissues, and overexpression of microRNA-9 directly inhibits the expression of *TNFAIP8*, leading to a decrease in gastric cancer cell proliferation in vitro and tumor growth in vivo, suggesting that TNFAIP8 is involved in gastric carcinogenesis and cancer progression [75]. Similarly, in gastric adenocarcinoma, higher expression of TNFAIP8 is associated with depth of invasion, lymph node metastasis, and poor prognosis [76]. In osteosarcoma (OS) tissues and OS cell lines, microRNA-99a expression is down-regulated compared with healthy bones tissues and normal osteoblastic cell lines respectively. The study demonstrated that overexpression of microRNA-99a decreases *TNFAIP8* expression in MG-63 and U2OS OS cells and leads to decreased cell viability in vitro. Introduction of *TNFAIP8* siRNA into OS MG-63 cells showed a reduction in tumor volume and weight of subcutaneous xenografted tumors in nude mice in vivo, suggesting that down-regulation of microRNA-99a and higher expression of TNFAIP8 promotes OS [77].

In esophageal squamous cell carcinoma (ESCC), overexpression of TNFAIP8 was found in 59.8% tumor specimens, and the 3-year lymphatic metastatic recurrence rate among TNFAIP8-overexpressing patients was significantly higher than in TNFAIP8 lower-expressing patients or TNFAIP8-negative patients [78]. Stable and transient knockdown of *TNFAIP8* in ESCC-derived cells (Eca109) and a second ESCC-derived cell line (KYSE150) decreases cell proliferation, motility, and invasion by the induction of cell apoptosis. The study suggests that higher expression of TNFAIP8 in ESCC is a potential biomarker for identification in pN0 ESCC patients [78]. The clinical relevance of TNFAIP8 was also determined in esophageal squamous cell carcinoma: higher expression of TNFAIP8 co-relates with TNM stage, tumor depth, lymph node metastasis, distant metastasis, lymphatic invasion, and venous invasion among ESCC patients, as well as with poor survival [79]. 

In epithelial ovarian cancers (EOC), the expression of TNFAIP8 is higher in platinum-resistant EOC compared with platinum-sensitive or normal ovaries. TNFAIP8 protein overexpression is correlated with optimal cytoreduction in EOC, whereas *TNFAIP8* mRNA expression is strongly associated with residual tumor size, suggesting that TNFAIP8 overexpression is an independent predictor of platinum resistance in EOC [80]. In another study using 202 epithelial ovarian cancer specimens, it was shown that expression of TNFAIP8 is associated with high histologic grade, large residual tumor size, recurrence and response to chemotherapy, revealing that TNFAIP8 may predict EOC metastasis and poor survival in epithelial ovarian cancers [81]. Similarly, in endometrial cancers, the expression of TNFAIP8 in tumor specimens is positively correlated with clinicopathologic factors such as higher histologic grade, deep myometrial invasion, lymphovascular space invasion, lymph node metastasis, and recurrence. TNFAIP8 expression strongly correlates with MMP9 and Ki-67 expression, suggesting that TNFAIP8 may be used as a prognostic marker for the recurrence of endometrial cancers [82]. TNFAIP8 expression in the survival subset of cervical cancer patients is significantly associated with resistance to cisplatin and nedaplatin, recurrence, and death from cervical cancer [83]. TNFAIP8 expression in pancreatic cancer tissue correlates with expression of epithelial growth factor receptor [84], and the study suggests that expression of TNFAIP8 in pancreatic cancer tissue is higher than normal pancreas tissue, indicating that TNFAIP8 promotes pancreatic cancer.

## 5. Conclusions and Perspectives

In the current review, we mainly focus on TNFAIP8 proteins and their roles in cancer biology. Since cytokine TNFα induces expression of TNFAIP8 in most cancers through activation of the NF-κB pathway, it is of interest to understand the biological role of TNFAIP8 proteins in the modulation of cancer cell signaling. The role of TNFAIP8 in cell survival or death appears to depend on the cellular context and the level of TNFα-mediated cellular inflammation. Since our analysis revealed that TNFAIP8 proteins may undergo several post-translational modifications such as glycosylation and phosphorylation, it is important to investigate how TNFα-mediated cellular inflammation affects the post-translation modification or activation of TNFAIP8 proteins in order to exert its effect on cancer cell survival/drug resistance. Numerous studies suggest that TNFAIP8 acts as an oncogenic molecule which induces drug resistance, cell proliferation, cell survival, cell metastasis, and autophagy in various types of cancer cells, by inhibiting apoptosis (Figure 5B). The molecular mechanisms by which TNFAIP8 modulates/promotes oncogenesis is still unknown and need further investigation.

Several studies and data from The Cancer Genomic Atlas (TCGA) suggest that the expression of TNFAIP8 protein in cancer tumors is generally higher compared with adjacent normal tissues. Furthermore, higher expression of TNFAIP8 in cancer tumors co-relates with low survival of patients compared with low TNFAIP8-expressing tumors. The TCGA data also suggest that *TNFAIP8* expression varies during development of different stages of cancer, for example, *TNFAIP8* expression in liver cancer stage I patients is similar to normal tissue expression, however, *TNFAIP8* expression significantly increased in liver cancer stage II and III, and then decreased in stage IV, suggesting that TNFAIP8 may be involved in the modulation of different stages of liver cancer. Designing effective TNFAIP8 inhibitors and controlling TNFAIP8 expression/stability, along with effective chemotherapeutic approaches, will be beneficial for controlling the development of various human cancers.

## Figures and Tables

**Figure 1 cells-08-00009-f001:**
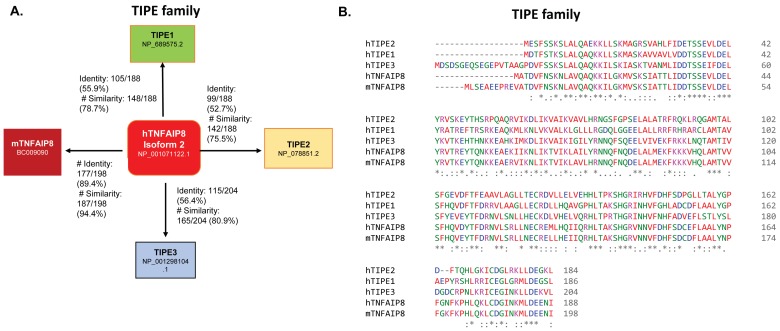
(**A**) Amino acid similarity and identity of TIPE family members, TIPE1, TIPE2, TIPE3, and mTNFAIP8, with humanTNFAIP8 (hTNFAIP8) isoform 2 are presented. The percentage of amino acid similarity and identity was determined by Emboss needle (http://www.ebi.ac.uk/Tools/psa/emboss_needle/ accessed on: 10 October 2018). (**B**) Amino acid sequence alignments of TIPE family members such as hTNFAIP8, mTNFAIP8, TIPE1, TIPE2, and TIPE3 were performed by using EMBL-EBI Clustal Omega software tools (https://www.ebi.ac.uk/Tools/msa/clustalo/, accessed on: 5 October 2018).

**Figure 2 cells-08-00009-f002:**
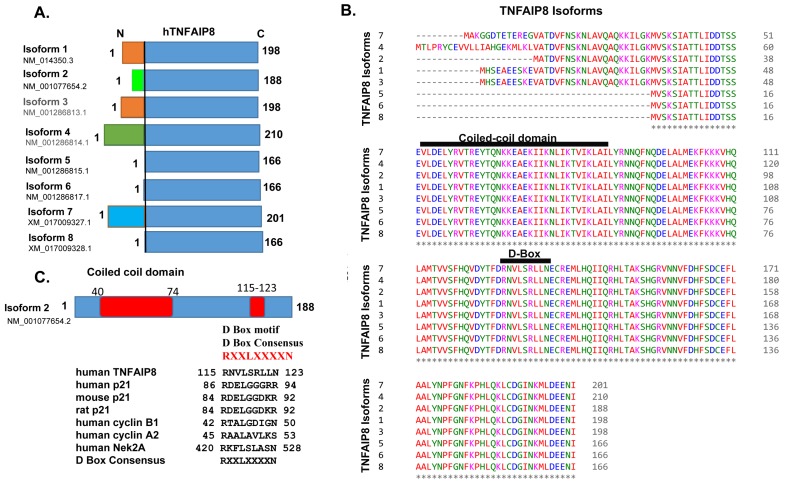
(**A**) Amino acid sequences of human TNFAIP8 (hTNFAIP8) protein isoforms are presented. The variable N-terminal region is shown in different colors. (**B**) Amino acid sequence alignments of hTNFAIP8 protein isoforms. The highly conserved coiled-coil structural motif and D-Box (destruction box) consensus are shown. (https://www.ebi.ac.uk/Tools/msa/clustalo/, accessed on: 5 October 2018). (**C**) Location of the coiled-coil domain and D-Box consensus on TNFAIP8 isoform 2 are presented. D-Box amino acid consensus of TNFAIP8 and other known D-Box motif-containing proteins are listed.

**Figure 3 cells-08-00009-f003:**
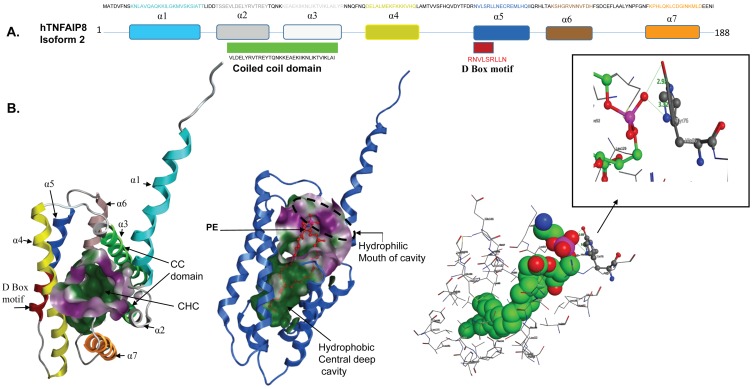
(**A**) The major seven α helices of human TNFAIP8 isoform 2 are presented in different colors and location of the coiled-coil domain and D-Box motif in helices are shown. (**B**) Rendering of the homology model of hTNFAIP8 isoform 2. Left panel: Colored ribbon diagram and molecular surface of the binding pocket, with pink representing the hydrophilic region and green the hydrophobic region. Middle Panel: Ribbon diagram with molecular surface of the binding pocket and the bound ligand (PE). Right Panel: Conserved pocket residues interacting with the ligand (PE). Human TNFAIP8-Tyr-76 and mouse TNFAIP8-His-86 form hydrogen bonds with the PE phosphate group, respectively. The molecular modeling software used for the graphic visualization was the Molecular Operating Environment (MOE 2016, Chemical Computing Group, Toronto, ON, Canada). The binding site surface was generated as molecular surface based on the alpha spheres generated in MOE. By MOE convention, pink color indicates hydrophilic surface region and green indicates hydrophobic region. PE—phosphatidylethanolamine. CC Domain—coiled-coil domain. CHC—central hydrophobic cavity.

**Figure 4 cells-08-00009-f004:**
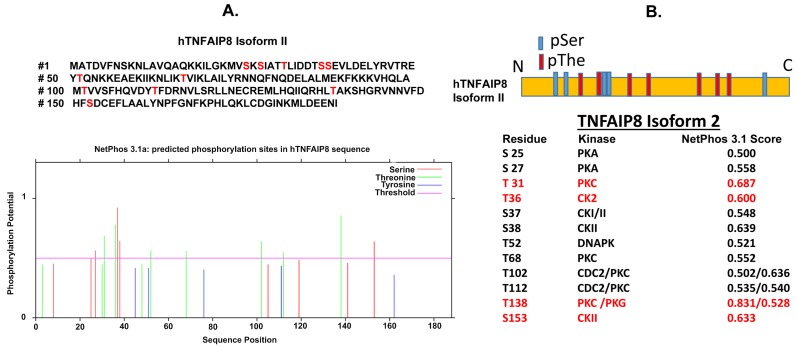
(**A**,**B**) Prediction of phosphorylation sites of TNFAIP8 were performed by using NetPhos server 3.1 (http://www.cbs.dtu.dk/services/NetPhos/, accessed on: 15 July 2018) and Prosite search (https://prosite.expasy.org/, accessed on: 10 July 2018) (**B**) Locations of predicted pSer and pThe phosphorylation sites of TNFAIP8 isoform 2 are presented (upper schematic). Phosphorylation sites, associated kinases, and NetPhos scores are presented (lower table). Both NetPhos and Pro-site server-predicted phosphorylation sites and kinases are shown in red color (lower table). pSer—phospho-Serine. pThe—phospho-threonine.

**Figure 5 cells-08-00009-f005:**
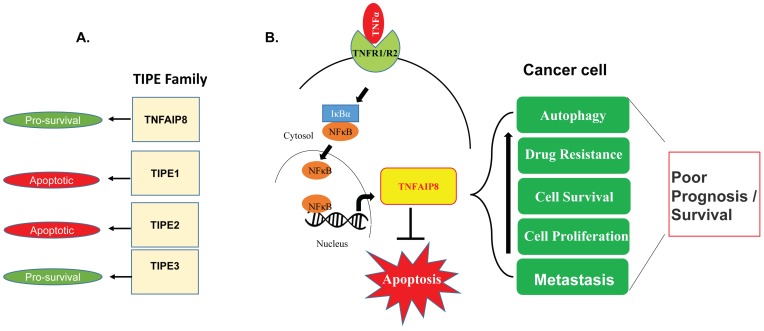
(**A**) The biological roles of TIPE family proteins in cell survival and cell death are suggested. (**B**) The schematic model represents the involvement of TNFAIP8 in the regulation of different cellular processes in cancer cells.

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
