# Peer review of "Oncogenic Role of Tumor Necrosis Factor α-Induced Protein 8 (TNFAIP8)"

_cells, 2018, doi:10.3390/cells8010009_

Round 1
Reviewer 1 Report
Overall, the manuscript is well-written. It is a good summary of TNFAIP8 biology and adds a well-needed review to the field. The biochemistry and associated figures exploring homology and isoforms is particularly valuable.
Author Response
Reviewer 1
Overall, the manuscript is well-written. It is a good summary of TNFAIP8 biology and adds a well-needed review to the field. The biochemistry and associated figures exploring homology and isoforms is particularly valuable.
Response: Thank for your comments.
Reviewer 2 Report
The authors have written an extensive review summarizing research findings related to the TNFAIP8 protein family, with particular focus on TNFAIP8. Overall, the review is solid, but the following points should be addressed:
1. Has it been shown experimentally that TNFalpha induces expression of ALL members of the TNFAIP8 gene family? (Lines 38-40, lines 55-56)
2. Can the authors provide references for their assertions in lines 42-45? Is this based on Figure 1?
3. The authors should provide the Ensembl designations for the TNFAIP8 splice forms (line 46).
4. Line 49. The authors should also reference the Lowe paper (reference 70) here.
5. In section 2, the authors should reference the following article, which addresses similar questions: https://journals.plos.org/plosone/article?id=10.1371/journal.pone.0179517
6. Lines 61, 63, 68, 70, 75, 273, 284, 294, elsewhere. When online tools are used, it is appropriate to identify the original reference so that one could be understand how the tools were developed and how they worked. Often, the purveyors of the tools request that this be done when using their work.
7. Line 73. High expression of protein or mRNA or both?
8. Line 135. “Protein Atlas” be capitalized?
9. Line 144. Where have the eight transcripts been reported?
10. Line 165. Where have the eight transcripts and five proteins been reported? Also, as stated earlier, it is appropriate to list the transcript names/designations.
11. Lines 205-07. Should we then assume that TNFAIP8, like TIPE2, does not contain a DED domain? Can this be stated explicitly?
12. In the TNFAIP8 interactions and signaling section, the authors should consider creating a summary figure.
13. Line 299. Did the authors use default parameter for scoring? What is the meaning of the score? Please include an article reference.
14. Line 320. Should “reported in the literature” be understood as “reviewed in the literature?”
15. Line 339. Where is TNFAIP8 found? There is reference to nuclear localization here. Isn’t it also a cytoplasmic protein? What about the other family members? Should that be stated earlier in the beginning of section 2.
16. Reference 70 was an interesting study that should be more broadly considered.
17. Line 371. H1299 cells lack a functional p53 due to partial deletion. Can the authors explain the context of these results given the p53 status of H1299 cells? It seems that there a misinterpretation of the data presented by Monteith et al.
18. Should the TNFAIP8-Hippo pathways shown in lung and liver cancers be better integrated in this review? Each story supports the other. There is a lack of connection in the way lines 379 -380 reads relative to lines 359-362.
19. Lines 430-441 are redundant and should be abridged. Lines 442-446 should be expanded further to share a viewpoint and provide a recommended direction.
Author Response
Reviewer 2
The authors have written an extensive review summarizing research findings related to the TNFAIP8 protein family, with particular focus on TNFAIP8. Overall, the review is solid, but the following points should be addressed:
Response: Thank for your comments and suggestions please see our point by point response to your comments below.
1. Has it been shown experimentally that TNFalpha induces expression of ALL members of the TNFAIP8 gene family? (Lines 38-40, lines 55-56)
Response: So far it is known from our study and others that TNF alpha induces expression of TNFAIP8 and TIPE2. It is not clear whether all members of the TIPE family are induced by TNF alpha. Text is modified in the revised manuscript.
2. Can the authors provide references for their assertions in lines 42-45? Is this based on Figure 1?
Response: It is based on Figure 1. (Figure 1A and URL have been added in the text)
3. The authors should provide the Ensembl designations for the TNFAIP8 splice forms (line 46).
Response: Ensembl URL provided.
4. Line 49. The authors should also reference the Lowe paper (reference 70) here.
Response: Reference added in the text.
5. In section 2, the authors should reference the following article, which addresses similar questions: https://journals.plos.org/plosone/article?id=10.1371/journal.pone.0179517
Response: Reference added in the text.
6. Lines 61, 63, 68, 70, 75, 273, 284, 294, elsewhere. When online tools are used, it is appropriate to identify the original reference so that one could be understand how the tools were developed and how they worked. Often, the purveyors of the tools request that this be done when using their work.
Response: As per Cells Journal reference style we provided URL and accessed date in the text.
7. Line 73. High expression of protein or mRNA or both?
Response: Clarified in the text.
8. Line 135. “Protein Atlas” be capitalized?
Response: Capitalized in the revised text.
9. Line 144. Where have the eight transcripts been reported?
Response: NCBI URL provided in the revised text.
10. Line 165. Where have the eight transcripts and five proteins been reported? Also, as stated earlier, it is appropriate to list the transcript names/designations.
Response: URL provided. Transcripts names and designations are presented in Figure 2A.
11. Lines 205-07. Should we then assume that TNFAIP8, like TIPE2, does not contain a DED domain? Can this be stated explicitly?
Response: Clarified in the revised text.
12. In the TNFAIP8 interactions and signaling section, the authors should consider creating a summary figure.
Response: Since these individual interactions are not validated by using IP/WB approach or other techniques, it is not appropriate to create a figure here; we will consider it in future.
13. Line 299. Did the authors use default parameter for scoring? What is the meaning of the score? Please include an article reference.
Response: CBS server, Net Phos and Prosite URLs are provided in the revised text. Generally, sites with scores greater than 0.5 are considered as significant sites for modification.
14. Line 320. Should “reported in the literature” be understood as “reviewed in the literature?”
Response: Text changed as per suggestion.
15. Line 339. Where is TNFAIP8 found? There is reference to nuclear localization here. Isn’t it also a cytoplasmic protein? What about the other family members? Should that be stated earlier in the beginning of section 2.
Response: Localization of TIPE family members including TNFAIP8 is mentioned in revised text.
16. Reference 70 was an interesting study that should be more broadly considered.
Response: Clearly and broadly considered and mentioned in the text.
17. Line 371. H1299 cells lack a functional p53 due to partial deletion. Can the authors explain the context of these results given the p53 status of H1299 cells? It seems that there a misinterpretation of the data presented by Monteith et al.
Response: Clear statements are made and text modified accordingly.
18. Should the TNFAIP8-Hippo pathways shown in lung and liver cancers be better integrated in this review? Each story supports the other. There is a lack of connection in the way lines 379 -380 reads relative to lines 359-362.
Response: Since the roles of TNFAIP8 in lung and liver cancers are separated by paragraphs, we modified the text to make a good connection about TNFAIP8-Hippo pathways in both types of cancer.
19. Lines 430-441 are redundant and should be abridged. Lines 442-446 should be expanded further to share a viewpoint and provide a recommended direction.
Response: As per reviewer’s suggestion we removed lines 430-441 and the whole section is modified. We also inserted the inputs suggested by Reviewer 3.
Reviewer 3 Report
In this manuscript the author summarized current knowledge of TNFAIP8 family proteins, covering the protein family introduction, expression, protein structure, modification, interaction and oncogenic function of this protein family. This review is comprehensively covered of the basal and updated information of researches on this potential oncogenic protein family. The manuscript is well written with clear logical flow.
Minor:
1. Current problems in TNFAIP8 study and translational potential TNFAIP8-related research may be discussed in Conclusions and Perspectives.
2. Typo: Nf-kβ should be NF-kB.
Author Response
Reviewer 3
In this manuscript the author summarized current knowledge of TNFAIP8 family proteins, covering the protein family introduction, expression, protein structure, modification, interaction and oncogenic function of this protein family. This review is comprehensively covered of the basal and updated information of researches on this potential oncogenic protein family. The manuscript is well written with clear logical flow.
Response: Thank for your comments and suggestions please see our point by point response below.
Minor:
1. Current problems in TNFAIP8 study and translational potential TNFAIP8-related research may be discussed in Conclusions and Perspectives.
Response: As per reviewer’s suggestion the whole section is modified and also inserted the inputs suggested by Reviewer 2
2. Typo: Nf-kβ should be NF-kB.
Response: Changed.